# The Silent Threat: Exploring the Ecological and Ecotoxicological Impacts of Chlorinated Aniline Derivatives and the Metabolites on the Aquatic Ecosystem

**Daniela Rebelo** [1,2,3,*] ⬤, **Sara C. Antunes** [2,3] ⬤ **and Sara Rodrigues** [2,3,*] ⬤

1   Abel Salazar Biomedical Sciences Institute (ICBAS), University of Porto, Rua de Jorge Viterbo Ferreira, 228, 4050-313 Porto, Portugal
2   Interdisciplinary Centre of Marine and Environmental Research (CIIMAR), Terminal de Cruzeiros do Porto de Leixões, Avenida General Norton de Matos S/N, 4550-208 Matosinhos, Portugal; scantunes@fc.up.pt
3   Department of Biology, Faculty of Sciences of the University of Porto (FCUP), Rua do Campo Alegre S/N, 4169-007 Porto, Portugal
*   Correspondence: up202210683@edu.icbas.up.pt (D.R.); sara.rodrigues@fc.up.pt (S.R.)

**Abstract:** The growing concern over the environmental impacts of industrial chemicals on aquatic ecosystems has prompted increased attention and regulation. Aromatic amines have drawn scrutiny due to their potential to disturb aquatic ecosystems. 4-chloroaniline and 3,4-dichloroaniline are chlorinated derivatives of aniline used as intermediates in the synthesis of pharmaceuticals, dyes, pesticides, cosmetics, and laboratory chemicals. While industrial applications are crucial, these compounds represent significant risks to aquatic environments. This article aims to shed light on aromatic amines' ecological and ecotoxicological impacts on aquatic ecosystems, given as examples 4-chloroaniline and 3,4-dichloroaniline, highlighting the need for stringent regulation and management to safeguard water resources. Moreover, these compounds are not included in the current Watch List of the Water Framework Directive, though there is already some information about aquatic ecotoxicity, which raises some concerns. This paper primarily focuses on the inherent environmental problem related to the proliferation and persistence of aromatic amines, particularly 4-chloroaniline and 3,4-dichloroaniline, in aquatic ecosystems. Although significant research underscores the hazardous effects of these compounds, the urgency of addressing this issue appears to be underestimated. As such, we underscore the necessity of advancing detection and mitigation efforts and implementing improved regulatory measures to safeguard the water bodies against these potential threats.

**Keywords:** xenobiotic; chloroaniline; water quality; aquatic ecotoxicology

## 1. Understanding Aromatic Amines

Aromatic amines are organic nitrogen compounds that consist of an amine coupled to an aromatic ring [1]. Many compounds belong to this class, such as phenylenediamines, toluidines, diaminotoluenes, naphtylamines, aminopyridines, aminopyrimidines, amino-quinolines, aminopurines, aminoacridines and anilines [2]. The simplest aromatic amine is named aniline or benzenamine (originated from benzene). In 1843, A. W. von Hofmann established those multiple substances identified before as "krystallin" (O. Unverdorben, 1826), "anilin" (C. J. Fritzche, 1841) and "benzidam" (N. Zinin, 1842), were a single nitrogenous base, that nowadays is named as aniline [3]. Aromatic anilines are commonly used in the production of dyes (e.g., Milliken, Netherlands), pharmaceuticals (e.g., DKSH Portugal, Unipessoal, Lda.), cosmetics (e.g., BASF Portuguesa, S.A., Portugal), laboratory chemicals (e.g., Lanxess, Germany), polymers (e.g., Covestro, Germany) and pesticides (e.g., Bondalti, Portugal) [4,5]. Considering aniline production, an increase from 2011 to 2021 at a rate of >1.1% per year was recorded. In 2021 it was estimated at around 605 thousand tons, and the biggest producers in the European Union (EU) were Belgium, Czechia, and Portugal,

with 293, 105, and 85 thousand tons exported, respectively. Total aniline consumption in the EU was around 905 thousand tons in 2021, where the most relevant countries were the Netherlands, Germany, Hungary, Spain, Portugal, the Czech Republic, and Belgium [5]. The country that presented a more significant increase in consumption from 2011 to 2021 was Belgium (+15.6%), comparatively, to the others mentioned [5].

Chloroanilines, such as 4-chloroaniline (4-CA) and 3,4-dichloroaniline (3,4-DCA), are used as building blocks in the synthesis of pharmaceutical products such as chlorhexidine and triclocarban, used in the production of antiseptic mouthwashes, deodorants, soaps, etc. [6,7]. These aromatic amines play a crucial role in the development of active pharmaceutical ingredients and are incorporated into the chemical structures of diverse drug classes, including analgesics, antipyretics, and antivirals [8]. The vibrant coloration in many dyes and pigments is also attributed to the presence of aromatic compounds such as chloroanilines [9]. These aromatic amines are used as precursors in the production of azo dyes, which are widely used in the textile, printing, and cosmetic industries [10]. They are also utilized in the synthesis of agrochemicals, such as herbicides and fungicides, which play a vital role in crop protection and enhancement [11]. The significance of chloroanilines in these industrial sectors underscores their economic importance and the extensive use of these aromatic amines in various manufacturing processes [4,5]. However, it is essential to consider the potential environmental impacts associated with their production, use, and disposal [12].

The synthesis of 4-CA can be achieved through different processes: (i) the reduction of 4-chloronitrobenzene using $SnCl_2$ as a reducing agent, producing the intermediates 4-chloronitrosobenzene and 2-amino-4-chlorophenol [13]; (ii) the reaction of 1,4-dichlorobenzene with ammonia, producing HCl as a byproduct [11]; (iii) the hydrogenation of chloronitrobenzene using a heterogeneous catalyst (e.g., PtZn), which can produce chloroanilines, bromoanilines, and iodoanilines as byproducts; and (iv) biodegradation of linuron (3-(3,4-dichlorophenyl)-1-methoxy-1-methylurea) [14,15]. Meanwhile, 3,4-DCA is a biodegradation product of various phenylurea, acylanilide, and phenylcarbamates herbicides, including: (i) linuron, producing N,O-dimethylhydroxylamine as a byproduct; (ii) 3,4-Dichloropropionanilide (propanil), producing propionate as a byproduct; and (iii) diuron (3-(3,4-Dichlorophenyl)-1,1-dimethylurea), producing a dimethylamine [16,17]. 3,4-DCA can also be degraded by *Acinetobacter baylyi* to form 4-CA [18]. 4-CA and 3,4-DCA can also be used for the synthesis of some herbicides (4-CA: monolinuron, monuron, and diflubenzuron; 3,4-DCA: linuron, diuron, and propanil) [19]. Linuron and diuron are considered as more relevant substances, being the two parent compounds and/or byproducts of 4-CA and 3,4-DCA where there is more knowledge relative to the ecotoxicological effects on various organisms, also making a connection with the toxicity of 4-CA and/or 3,4-DCA [14–17,19].

The chemical processes involved in the synthesis of 4-CA and 3,4-DCA certainly have associated challenges and potential environmental problems. Chemical waste can include spent catalysts, unreacted starting materials, and byproducts that are not part of the desired final product, degrade slowly, bioaccumulate, contaminate the water, and impact soil health [12]. High concentrations of compounds used in the synthesis of 4-CA and 3,4-DCA can harm ecosystems through air pollution and soil damage, for example, via eutrophication, when excess nutrients are introduced in the soil, which can disrupt the balance of nutrients, negatively impacting the growth and health of plants, or altering the structure and functioning of soil microbial communities [20,21]. This can have cascading effects and consequently lead to biodiversity loss [22]. Herbicides based on 4-CA and 3,4-DCA can leach into groundwater and induce poor health conditions in the individuals exposed. Some of these compounds disrupt hormone production and are potentially genotoxic, mutagenic, and carcinogenic [23,24]. On the other hand, the acidification of water bodies caused by the substances involved in the synthesis of 4-CA and 3,4-DCA is also a concern. As the pH of water decreases, aromatic amines properties and bioavailability can be affected through their protonation, leading to a decrease in their solubility and

potentially altering the speciation and transformation of these compounds, which can impact their toxicity and persistence in the environment [25].

These risks necessitate careful waste management practices to prevent contamination of water and soil and the release of harmful substances into the atmosphere. Moreover, workers' exposure to these chemicals must be carefully managed to prevent health risks. However, the specific risks and waste products depend on the methods used and cannot be universally defined, depending on various factors, including the starting materials, intended applications, specific methods, and conditions used in the synthesis process [26–29]. Due to the lack of information concerning the environmental presence and impact of aromatic amines (compounds with high and diverse use) on aquatic ecosystems, this opinion paper intends to review the information available concerning these two compounds (4-CA and 3,4-DCA), identifying their environmental effects on aquatic organisms and measures to be taken into consideration to safeguard the environment.

## 2. Environmental Presence

The entrance of 4-CA and 3,4-DCA into the environment can occur through various pathways, including wastewater discharges, accidental spills, and atmospheric deposition [30]. These pathways can lead to the contamination of water bodies and pose risks to aquatic organisms. Even after undergoing treatment, trace amounts of these chemicals can still be present in wastewater-discharged effluents, potentially affecting downstream ecosystems [31]. In Table 1, environmentally relevant concentrations (ERC) of 4-CA and 3,4-DCA, already detected worldwide and from different aquatic matrices, are evidenced [32–40]. The information resumed in Table 1 is a result of a literature review (1988–2022) where specific keywords were used: aromatic amine, chloroaniline, 4-chloroaniline, 4-CA, p-chloroaniline, 3,4-dichloroaniline, 3,4-DCA, drinking water, wastewater influent, wastewater effluent, river, superficial water "https://scholar.google.com/ (accessed on 15 June 2023)".

**Table 1.** Literature review of environmental concentrations (μg/L) of 4-chloroaniline (4-CA) and 3,4-dichloroaniline (3,4-DCA) recorded in drinking water treatment plant (DWTP) influents, wastewater treatment plant (WWTP) influents and effluents, river superficial water, and groundwater.

| Compound | Type of Water | Site | μg/L | Reference |
|---|---|---|---|---|
| 4-CA | DWTP influent | Spain | <0.00006 | [32] |
| | | | 0.00022–0.0055 | |
| | WWTP influent and superficial water | Germany | 1.1–67 | [33] |
| | WWTP influent, effluent, and superficial water | Yangzhong, Yangtze River and tributaries, China | 0.0382–2.427 | [34] |
| | River—Superficial water | Zonguldak, Turkey | 0.00066–0.00082 | [35] |
| 3,4-DCA | DWTP influent | Spain | 0.0006–0.0025 | [32] |
| | | | 0.00092–0.0059 | |
| | | | 0.0012–0.0051 | |
| | WWTP influent and superficial water | Germany | 1.8–3.3 | [33] |
| | WWTP effluent | Tres Rios Wetlands—Hayfield Inlet, USA | 0.15 | [36] |
| | | | 0.3 | |
| | | Tres Rios Wetlands—Hayfield 2 Outlet, USA | 0.34 | |
| | | | 0.47 | |
| | | Homestead, Florida, USA | 0.095 | [37] |

**Table 1.** *Cont.*

| Compound | Type of Water | Site | µg/L | Reference |
|---|---|---|---|---|
| 3,4-DCA | River—superficial water | Zonguldak, Turkey | 0.00113–0.00194 | [35] |
| | | Almonda River upstream, Portugal | ≤6.82 | [38] |
| | | Almonda River downstream, Portugal | ≤20.19 | |
| | | Tres Rios Wetlands, Gila River, USA | 0.17 | [36] |
| | | Homestead, Florida, USA | 0.17 | [37] |
| | | Homestead, Florida—Monitoring well 1, USA | 0.11 | |
| | | | 0.086 | |
| | | Homestead, Florida—Monitoring well 2, USA | 0.2 | |
| | | | 0.14 | |
| | | Homestead, Florida—Monitoring well 3, USA | 0.15 | |
| | | | 0.092 | |
| | River—groundwater | Mondego River drainage basin, Portugal | ≤13.360 | [39] |
| | | Sado River basin, Portugal | ≤0.71 | [40] |

In addition, accidental spills during the transportation, storage, or handling of chloroanilines can lead to direct contamination of water bodies and, consequently, to immediate exposure in nearby aquatic ecosystems, with potential toxic effects on organisms [41]. These compounds can also be released into the atmosphere during industrial operations, combustion processes, or volatilization from contaminated surfaces. Once in the air, chloroanilines can be transported over long distances and eventually be deposited into land or water surfaces through precipitation or dry deposition, contaminating remote water bodies, even in areas far away from the pollution source [42]. 4-CA and 3,4-DCA can undergo diverse transformation processes (e.g., reductive dichlorination and microbial mineralization), may exhibit relative stability, adsorb to suspended particles in water bodies, and have leaching potential [43,44].

These compounds are considered persistent in the environment regarding the chemical properties in water. 4-CA presents a boiling point of 232 °C, a vapor pressure of 0.015 mm Hg at 25 °C [11], it can be volatilized (35.7 days half-life in rivers), photo oxidized in surface water (1 to 3 h half-life with low organic matter) and is biodegraded (several days to months half-life) [45]. 3,4-DCA showed a boiling point of 272 °C, vapor pressure of 1 mm Hg at 81 °C [46], and no evidence of hydrolysis, volatilization, and biodegradation. However, it undertakes photolysis in surface water (18-day half-life) and can bioaccumulate in groundwater sediments due to its very slow degradation rate (1000 days half-life) [47].

## 3. Impact on Aquatic Organisms

The freshwater biodiversity threat refers to the substantial decline in species diversity and abundance within freshwater ecosystems, including rivers, lakes, and wetlands [48]. This occurs due to multiple factors, including habitat loss, water pollution, invasive species, overfishing, climate change, altered water flow, and insufficient conservation efforts [49]. The repercussions of this crisis extend widely, affecting ecosystem services and human livelihoods alike [50]. Successfully addressing the environmental impacts is important for a comprehensive approach involving habitat protection, pollution management, sustainable water practices, invasive species control, and international cooperation [51].

Aromatic amines, despite being utilized in industrial and commercial sectors, present a notable hazard to freshwater ecosystems and can exacerbate the ongoing biodiversity threat. Entering water bodies through diverse pathways, these compounds endure, resulting in water pollution [52]. Exhibiting toxicity to aquatic organisms induces species depletion, degrades habitats, and fosters bioaccumulation [12]. Regarding these effects, it is urgent

to implement measures to mitigate the impact of aromatic amines on the preservation of freshwater biodiversity.

As previously reported, 4-CA and 3,4-DCA have been detected at the ng/L to µg/L in the aquatic compartment (Table 1) and are classified as persistent in aquatic environments, leading to potential bioaccumulation in organisms across the food web [52]. Some ecotoxicological studies have demonstrated that these compounds can disrupt behavior, growth, reproduction, and development, as well as cause sub-individual alterations (Table 2) [38,53–69].

**Table 2.** Literature review of ecotoxicological effects in different species after exposure to 4-chloroaniline (4-CA), 3,4-dichloroaniline (3,4-DCA), linuron and diuron (parent compounds and/or byproducts of 4-CA and 3,4-DCA).

| Compound | Organisms | Updated Species/Strains | Ecotoxicological Effects | µg/L | Reference |
|---|---|---|---|---|---|
| 4-CA | Bacteria | *Aliivibrio fischeri* (Beijerinck, 1889) | $EC_{50}$ (15 min) (bioluminescence inhibition) | 3760–34,300 | [54] |
| | Algae | *Raphidocelis subcapitata* (Korshikov, 1953) | $EC_{50}$ (72 h) (growth inhibition) | 1500 | |
| | Invertebrates | *Daphnia magna* (Straus, 1820) | $EC_{50}$ (48 h) (immobilization) | 100–310 | |
| | | *Ruditapes philippinarum* (Payraudeau, 1826) | Oxidative stress, oxidative damage, and genotoxicity (15 days) | ≥500 | [55] |
| | Vertebrates | *Danio rerio* (F. Hamilton, 1822) | $LC_{50}$ (96 h) (death of juveniles) | 30,700–46,000 | [54] |
| | Mammals | Male F344/N rats *Rattus norvegicus* (Berkenhout, 1769) | Tumors in the spleen (week 74) [1],* | ≥15,300 µg/kg/day | [56] |
| | | Male B6C3F1 *Mus musculus* (Linnaeus, 1758) | Tumors in the spleen, liver and kidney (male: week 72; female: week 89) [2],* | ≥514,000 µg/kg/day | |
| | | Female B6C3F1 *Mus musculus* (Linnaeus, 1758) | | ≥557,000 µg/kg/day | |
| 3,4-DCA | Bacteria | *Aliivibrio fischeri* | $EC_{50}$ (30 min) (bioluminescence inhibition) | 610–1500 | [38,57] |
| | Algae | *Scenedesmus obliquus* (Kützing, 1833) | $EC_{50}$ (96 h) (growth inhibition) | 7940 | [58] |
| | | *Chlorella pyrenoidosa* (Chick, 1903) | $EC_{50}$ (72 h) (growth inhibition) | 8440 | |
| | Invertebrates | *Daphnia magna* | $EC_{50}$ (48 h) (immobilization) | 310–226,000 | [59,60] |
| | | *Magallana gigas* (Thunberg, 1793) | Genotoxicity (6 h) | ≥0.05 [#] | [61] |
| | Vertebrates | *Danio rerio* | $LC_{50}$ (96 h) (death of juveniles) | 3200 | [62] |
| | | *Carassius auratus* (Linnaeus, 1758) | Oxidative stress and oxidative damage (15 days) | ≥200 | [60] |
| | | *Oryzias javanicus* (Bleeker, 1854) | Decrease in fecundity in females (21 days) [3] | ≥250 [#] | [53] |
| | | *Oreochromis niloticus* (Linnaeus, 1758) | Antiandrogenic activity in males (25 days) | ≥0.2 [#] | [63] |

**Table 2.** *Cont.*

| Compound | Organisms | Updated Species/Strains | Ecotoxicological Effects | µg/L | Reference |
|---|---|---|---|---|---|
| Linuron (parent compound and/or byproduct of 4-CA and 3,4-DCA) | Vertebrates | *Oncorhynchus mykiss* (Walbaum, 1792) | Histopathological damage in the liver and gills and oxidative stress (21 days) | ≥30 | [64] |
| Diuron (parent compound and/or byproduct of 3,4-DCA) | Plants | *Elodea canadensis* (Michaux, 1803) *Myriophyllum spicatum* (Linnaeus, 1753) *Potamogeton lucens* (Linnaeus, 1753) | Phytotoxic (5 weeks) | ≥0.2 | [65] |
| | Invertebrates | *Magallana gigas* | Genotoxic, embryotoxic (24 h) | ≥0.05 | [61,66] |
| | | | Immunotoxic (4 weeks) | ≥0.3 | [67] |
| | Vertebrates | *Oreochromis niloticus* | Endocrine disruptor (25 days) | ≥0.2 | [63] |
| | Mammals | Male Wistar rat and human urothelial cell | Carcinogenic, mutagenic, cytotoxic and neurotoxic alterations. Disruption of endocrine, cardiovascular and respiratory functions (3 days) | 0.05–0.5 | [68,69] |

[1] Fibromas, fibrosarcomas, hemangiosarcomas, osteossarcomas or sarcomas. [2] Hemangiosarcomas or hemangiomas. [3] Reduction in spawning rate, fertilization, gonadosomatic index, and disruption in oocyte development. * Exposure via feeding. # Environmentally relevant concentrations (ERC) (Table 1).

Considering this literature review (Table 2), behavioral changes (e.g., feeding patterns, locomotion, predator-prey interactions, and avoidance), as well as the disruption of growth, reproduction, and development of aquatic organisms (e.g., reduced growth rates, gamete production, fertilization, and impaired development) after exposure to xenobiotics, can impact the ability of individuals to find food, evade predators, reduce fitness and reproductive capacity [64]. This can ultimately affect survival and reproductive success, having long-term implications for aquatic ecosystems' population dynamics and health [70]. However, the literature has limited information on the ecotoxicological impacts of 4-CA and 3,4-DCA at environmentally relevant concentrations.

## 4. Status in Water Framework Directive

The Water Framework Directive (WFD, adopted in 2000) is a legislation in the EU focused on water policy that aims to achieve and maintain all water bodies with a good ecological and chemical status. Both 4-CA and 3,4-DCA compounds are listed as candidates for the 4th Watch List under the WFD [71]. 4-CA and 3,4-DCA were originally selected as Priority 1, but after Member States and Stakeholder experts' input regarding uncertainties considering the predicted no-effect concentration (PNEC), Joint Research Center experts chose to gather more data before including them in the Watch List, being moved into the Priority 2 category [71]. Priority 1 and Priority 2 substances contain the "most suitable candidates for inclusion in the next Watch List" and "almost suitable candidates for the next Watch List, but for which a reliable PNEC or analytical method is missing". However, these two categories include compounds that may represent a risk in the aquatic compartment, having limited or low-quality monitoring data in the EU for risk assessment.

## 5. Environmental Concerns about Aromatic Amines

Detecting and measuring chemicals such as aromatic amines in the environment poses several challenges due to the low concentrations (ng/L to μg/L), complex matrices, and the need for sensitive and selective analytical quantification techniques. However, advancements in analytical methods, through the implementation of specific techniques such as a derivatization step prior to gas chromatography (GC), extraction processes (e.g., liquid–liquid extraction or solid-phase extraction) followed by high-performance liquid chromatography (HPLC), capillary zone electrophoresis (CZE) and spectrophotometric methods, can provide improved capabilities for the detection of these specific compounds in natural waters [72].

Moreover, it is crucial to address the issue of contaminated water sources and implement appropriate measures to reduce the exposure of aquatic individuals to aromatic amines. The specific mechanism of action of aromatic amines on aquatic environments and organisms cannot be understated. The evidence gathered from scientific studies, such as the potential for long-range transport and the persistence of aromatic amines in the environment, highlights the need for stringent regulation and management practices to prevent the release of these chemicals into water bodies [73]. Efforts should focus on reducing the use with the promotion of alternative substances, for instance, non-halogenated anilines (e.g., methylanilines, nitroanilines, or methoxyanilines) or less toxic aromatic compounds, as intermediates in the industry where chloroanilines are currently used [74]. On the other hand, it is important to implement effective wastewater treatment processes that have been referred to in the literature for aniline removal, such as Electro-Fenton® and peroxy-coagulation processes using a flow reactor, implementing the use of a sequencing batch reactor, zero-valent ion coupled with hydrogen peroxide, among others [75–77]. These methods can reduce the concentration of aromatic amines in wastewater treatment plant (WWTP) effluents, mitigating the impact on the ecosystems. Striving for a sustainable future demands proactive measures to protect water resources and the delicate ecosystems they support.

## 6. Future Developments

The big challenge for the future is to reduce the impacts of anthropogenic activities on natural resources and protect ecosystems to achieve an ecologically sustainable environment. Given the wide use of these compounds in industry for different purposes, as well as the wide dissemination to the environment via elimination processes or as a result of degradation products, it adds greater concern to the environmental presence. Additionally, it is empirical to anticipate the environmental impacts of aromatic amines before their undue occurrence, employing more effective analytical methodologies to study the interrelation between the health of organisms, even at sub-individual levels (more sensitive and early warning responses) and the ecosystem. A response at sub-individual levels to stressors, i.e., at the cellular level, such as perturbations of oxidative metabolism, after exposure to these compounds, can potentially initiate outcomes at individual levels, manifesting as changes in behavior and reproductive irregularities or even death. In such scenarios, a cascade effect can incite implications at the population level, such as altered abundance, species decay, or population dynamics. Consequently, sequential effects on the community level may be detected, such as disruption in food chains, alterations in community composition, or changes in biodiversity. Lastly, these influences can scale up to the ecosystem level, inducing functional changes such as nutrient cycling disturbance, a shift in energy flow, or alteration in ecosystem services [78]. Therefore, effective anticipatory measures require complex and comprehensive prospective risk assessments, taking into consideration this complex web of interactions.

Although in the Water Framework Directive, 4-CA and 3,4-DCA are still not included as compounds for monitorization to ensure a good ecological water status, they are currently detected between the ng/L and μg/L range, as evidenced in Table 1. These environmentally relevant concentrations are also the threshold values to be taken into consideration for risk

assessment and policies implemented for these compounds. It is also empirical to consider that at these concentrations, and as evidenced above, several harmful ecotoxicological effects are reported at the sub-individual and individual level in several organisms and populations, including in the aquatic compartment. Therefore, these findings cannot be disregarded.

Considering that WWTP can only partially remove these compounds from wastewater [31] and the production and consumption of aromatic amines has increased in recent years [5], this will likely lead to an increase in the concentrations in surface waters. In this case, an environmental risk assessment concerning aromatic amines becomes crucial for the implementation of policies to restrict their usage and diffusion through the aquatic environment. Particularly for compounds with low rates of degradation and directly applied in the environment, that can have a more detrimental impact on aquatic organisms.

Due to its wide use, the diverse possibilities of reaching the environment, persistence, analytical methods of analysis (often inappropriate), and the lack of scientific evidence on the real risks to aquatic ecosystems (considering the above factors) reinforce the silent threat that aromatic amines, mainly 4-CA and 3,4-DCA, represent for a sustainable environment balanced with anthropogenic needs.

**Author Contributions:** Writing—original draft preparation, D.R.; writing—review and editing, D.R., S.C.A. and S.R.; supervision—S.C.A. and S.R. All authors have read and agreed to the published version of the manuscript.

**Funding:** The research conducted on this topic was funded by the Foundation for Science and Technology and by the Strategic Program UIDB/04423/2020 and UIDP/04423/2020. Sara Antunes and Sara Rodrigues are hired through the Regulamento do Emprego Científico e Tecnológico—RJEC from the FCT program (CEECIND/01756/2017 and 2020.00464.CEECIND, respectively). Daniela Rebelo is supported by FCT Ph.D. grants (2022.13777.BD).

**Institutional Review Board Statement:** Not applicable.

**Informed Consent Statement:** Not applicable.

**Data Availability Statement:** No new data were created or analyzed in this study. Data sharing is not applicable to this article.

**Conflicts of Interest:** The authors declare no conflict of interest.

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
