# Peer review of "The Silent Threat: Exploring the Ecological and Ecotoxicological Impacts of Chlorinated Aniline Derivatives and the Metabolites on the Aquatic Ecosystem"

_jox, doi:10.3390/jox13040038_

Round 1

Reviewer 1 Report

Comments and recommendations for authors

The title should refer strictly to the two compounds for which arguments of high risk of toxicity to aquatic organisms are made, i.e. chlorinated aniline derivatives and some of their metabolites.

From the section on “Understanding aromatic amine”, related to the properties and also the economic importance of these aromatic compounds, it appears that in addition to their toxic effects on the environment (through the use of dyes and pesticides), there are also their beneficial effects in the pharmaceutical industry or for medicine (pharmaceuticals and medicines). Therefore, in section “Conclusions”, measures should be proposed to control the use of only those substances that have toxic effects on the environment. We cannot limit the production of certain drugs or pharmaceuticals.

In the section “Impact on aquatic organisms”, I recommend a table showing the effects of the two compounds on different aquatic organisms (bacteria, plants, invertebrates, mammals): EC50%, LC50%, growth inhibition of plants, etc. The presentation in a table would better organize the information, which is presented in a block and is difficult to correlate with the concentrations of compounds in the environment (Table 1). From 26 references of this section, very few recent papers (3 papers in the last 5 years, 8 papers in the last 10 years) have been mentioned for the two compounds analysed. I have exemplified some references that are not justified, not related to the subject: 66, 67 and 71.

Citation 66 refers to studies on rats, in which the diet had a high intake of diuron; toxicity due to DCA is not clearly specified

Citation 66 is not related to the explanation in lines: 174 and 175

Citation 67 is not complete: - missing source (SSRN) and date of publication (June 2023) and does not relate to the explanation in the manuscript in lines 174 and 175.

Citation 69 - no direct link to the effects mentioned in the manuscript. Same for citation 71.

In lines 181-187 - citation 72 is not related to the diuron metabolites mentioned in that section. 

Author Response

Thank you very much for taking the time to review this manuscript. Please find the detailed responses and the corresponding revisions/corrections highlighted in yellow in the re-submitted files.

Reviewer 2 Report

Dear Authors,

I read your paper titled" The silent threat: exploring the ecological and ecotoxicological impacts of aromatic amines on the aquatic ecosystem ". I believe that it [the manuscript] provides a good structure, but the sections should provide an in depth view on the problem. The manuscript fails in this aspect. Please find below specific suggestions.  

1. Understanding aromatic amines

Line 51-52. Reference is needed.

Line 55. Please change to "deodorants, soaps etc. [6,7]."

Line 64-67. Reference is needed.

General comment. Authors should show the novelty of the manuscript in the first section. Some questions are: is there a similar study or the study is the first to provide an opinion on the problem?  

2. Environmental presence

General comment. Which methods were used to review literature in Table 1? Source and keywords should be mentioned.  

3. Impact on Aquatic Organisms

General comment. This section is a great point of your work. However, an in depth review is needed. This includes an exhaustive search for references a better connection with the real problem of our biodiversity (e.g., freshwater biodiversity crisis). How aromatic amines may exacerbate the problem? This should be explored. 

General comment. Please include the name of the author of the species for animal species mentioned in your text.

Author Response

Thank you very much for taking the time to review this manuscript. Please find the detailed responses and the corresponding revisions/corrections highlighted in green in the re-submitted files.

Reviewer 3 Report

In the manuscript entitled "The silent threat: exploring the ecological and ecotoxicological impacts of aromatic amines on the aquatic ecosystem " (toxics-2563842), submitted to Journal of Xenobiotics, you provided an overview of chlorinated aromatic amines (4-chloranilie and 3,4-dichloranilie) environmental presence and concerns, impact on aquatic ecosystems, and recommendations towards development analytical methodologies to study their interactions with aquatic organisms enabling integrative environmental risk assessment approach of aromatic amines.

The manuscript is well written and clearly presents the topic mentioned above. Thereby, I believe that the manuscript deserves to be published in Journal of Xenobiotics in present form

Author Response

Thank you very much for taking the time to review this manuscript. We appreciate the positive feedback and hope to continue to fulfill your expectations in the revised MS version.

Round 2

Reviewer 1 Report

The manuscript has been completed according to the recommendations made and can be published.